# Analysis Strategies for MHz XPCS at the European XFEL

Francesco Dallari [1,*,†], Mario Reiser [2,*,†], Irina Lokteva [1,3], Avni Jain [1,3,‡], Johannes Möller [4], Markus Scholz [4], Anders Madsen [4], Gerhard Grübel [3,5], Fivos Perakis [2] and Felix Lehmkühler [3,5]

1   Deutsches Elektronen-Synchrotron DESY, Notkestr. 85, 22607 Hamburg, Germany;
    irina.lokteva@desy.de (I.L.); avnij10@gmail.com (A.J.)
2   AlbaNova University Center, Department of Physics, Stockholm University, SE-10691 Stockholm, Sweden;
    f.perakis@fysik.su.se
3   The Hamburg Centre for Ultrafast Imaging, Luruper Chaussee 149, 22761 Hamburg, Germany;
    gerhard.gruebel@desy.de (G.G.); felix.lehmkuehler@desy.de (F.L.)
4   European X-ray Free-Electron Laser Facility, Holzkoppel 4, 22869 Schenefeld, Germany;
    johannes.moeller@xfel.eu (J.M.); markus.scholz@xfel.eu (M.S.); anders.madsen@xfel.eu (A.M.)
5   Center for Molecular Water Science CMWS, Deutsches Elektronen-Synchrotron DESY, Notkestr. 85,
    22607 Hamburg, Germany
*   Correspondence: francesco.dallari@desy.de (F.D.); mario.reiser@fysik.su.se (M.R.)
†   These authors contributed equally to this work.
‡   Current address: Blue Yonder GmbH, Oberbaumbrücke 1, 20457 Hamburg, Germany.

**Abstract:** The nanometer length-scale holds precious information on several dynamical processes that develop from picoseconds to seconds. In the past decades, X-ray scattering techniques have been developed to probe the dynamics at such length-scales on either ultrafast (sub-nanosecond) or slow ((milli-)second) time scales. With the start of operation of the European XFEL, thanks to the MHz repetition rate of its X-ray pulses, even the intermediate $\mu$s range have become accessible. Measuring dynamics on such fast timescales requires the development of new technologies such as the Adaptive Gain Integrating Pixel Detector (AGIPD). $\mu$s-XPCS is a promising technique to answer many scientific questions regarding microscopic structural dynamics, especially for soft condensed matter systems. However, obtaining reliable results with complex detectors at free-electron laser facilities is challenging and requires more sophisticated analysis methods compared to experiments at storage rings. Here, we discuss challenges and possible solutions to perform XPCS experiments with the AGIPD at European XFEL; in particular, at the Materials Imaging and Dynamics (MID) instrument. We present our data analysis pipeline and benchmark the results obtained at the MID instrument with a well-known sample composed by silica nanoparticles dispersed in water.

**Keywords:** XPCS; European XFEL; data analysis; dynamics; diffusion

## 1. Introduction

Free-electron laser facilities in the hard X-ray regime (XFELs) bear the potential for studying molecular dynamics utilizing time-domain methods such as X-ray photon correlation spectroscopy (XPCS) and the related technique X-ray speckle visibility spectroscopy (XSVS). Both techniques are based on coherent X-rays and enable probing dynamics between femtoseconds and several hours. These techniques have been developed at synchrotron radiation sources since the 1990s [1–4]. Applications cover a broad range of materials and scientific questions, such as diffusion dynamics in soft matter, glass transition, and gelation, as well as domain-wall dynamics; see [5] for a recent overview.

In XPCS experiments, the sample dynamics are studied by acquiring a series of coherent diffraction patterns, so-called speckle patterns. The intensity fluctuations of the speckles reflect the change of the spatial arrangement of the sample where the length scale is selected by choosing a particular momentum transfer $q \equiv |\mathbf{q}| = \frac{4\pi}{\lambda} \sin(\theta/2)$, with wavelength $\lambda$ and scattering angle $\theta$. The intensities $I(q, \tau)$ and $I(q, \tau + t)$ at two different times with a lag time of $t$ are recorded to calculate intensity–intensity correlation functions

$$g_2(q,t) = \frac{\langle I(q,\tau) I(q,\tau+t) \rangle}{\langle I(q,\tau) \rangle^2} \,. \tag{1}$$

The average denoted by $\langle \dots \rangle$ is performed over both detector pixels with equivalent $q$-values and all times $\tau$. The correlation function, $g_2$, can be expressed by the intermediate scattering function, $f(q,t)$, that describes the temporal evolution of the spatial arrangement of the sample. Via the *Siegert-relation*, Equation (1) can be written as

$$g_2(q,t) = 1 + \beta |f(q,t)|^2, \tag{2}$$

with the speckle contrast $\beta$ that is determined by the coherence properties of the beamline [5]. While XPCS at storage-ring sources typically covers dynamics in the range between hours down to (sub-)millisecond time scales, its main application at XFEL facilities are fast time scales in the femto- to nanosecond domain [6–8], using either double-pulse approaches via split-and-delay devices or modification of the X-ray pulse length between a few to about 100 fs. The apparent gap of time scales between nano- and milliseconds originates from the limitations of the time resolution of two-dimensional detectors. Modern photon-counting detectors used for XPCS at storage-ring facilities reach kHz repetition rates [5] and thus define the experimental limit. Recently, this limit has been reduced to microsecond dynamics using new detector generations [9,10]. However, these experiments are nowadays mainly limited to model sample systems and small-angle scattering geometries, where the scattered intensity is high enough to obtain a decent statistical accuracy. In contrast, the high intensities obtained from single FEL pulses demand the use of integrating detectors. Therefore, different detectors have been developed at FEL sources, such as the CSPAD and ePix at LCLS [11–13] and the MPCCD [14] at SACLA. These detectors match the repetition rate of the FEL in the range of 30 Hz to 120 Hz. The special pulse scheme of the European XFEL, where X-ray pulses are generated every 0.1 s in trains consisting of up to 2700 pulses at repetition rates up to 4.5 MHz, require a more demanding detector development. One of the detectors developed for the European XFEL is the Adaptive Gain Integrating Pixel Detector (AGIPD) [15]. With its pixel size of 200 μm and a repetition rate up to 6.5 MHz, it promises to perform routinely XPCS experiments on (sub-)microsecond time scale. Due to the high intensity of each pulse, experiments on molecular length scales are also in reach. AGIPD is designed to be used by very different scientific communities. Each detector pixel has three gain stages (high, medium, low gain) that are switched automatically depending on the detected intensity. The ability to detect bursts of extremely intense X-rays is its main feature, but under the appropriate conditions it can achieve single photon sensitivities in high-gain mode [16].

Recently, we demonstrated the measurement of sub-microsecond dynamics in soft matter systems using XPCS at the European XFEL [17,18]. We used the AGIPD that is installed at both the SPB/SFX [19] and the MID instrument, the latter is the preferred instrument for XPCS and XSVS experiments due to the long sample-detector distance of up to 8 m [20]. The tools that have been developed to analyze XPCS data at synchrotron radiation sources with commercial detectors like the Eiger series [21,22] are not capable of analyzing the complex datasets produced by the AGIPD that can reach volumes of up to Petabytes per week. The burst mode operation scheme of the AGIPD was found to yield several sources of noise and correlations within the noise that can produce severe artifacts in the XPCS analysis. Therefore, specialized analysis strategies and methods have to be developed, which account for these additional effects.

In this work we present the XPCS data pipeline developed by us and explain the various steps that are required to perform an XPCS experiment with the AGIPD at the European XFEL instruments. We distinguish between two different aspects of the data treatment. First, we will explain how the raw data should be calibrated to provide the best possible data quality of single speckle images. Second, we will discuss how the standard XPCS analysis (cf. Equation (1)) needs to be modified to correct detector burst mode characteristics. Eventually, we compare different data calibration and analysis methods

to benchmark the different approaches. A summary of our data pipeline is sketched in Figure 1. A detailed explanation of the individual steps follows in the next sections.

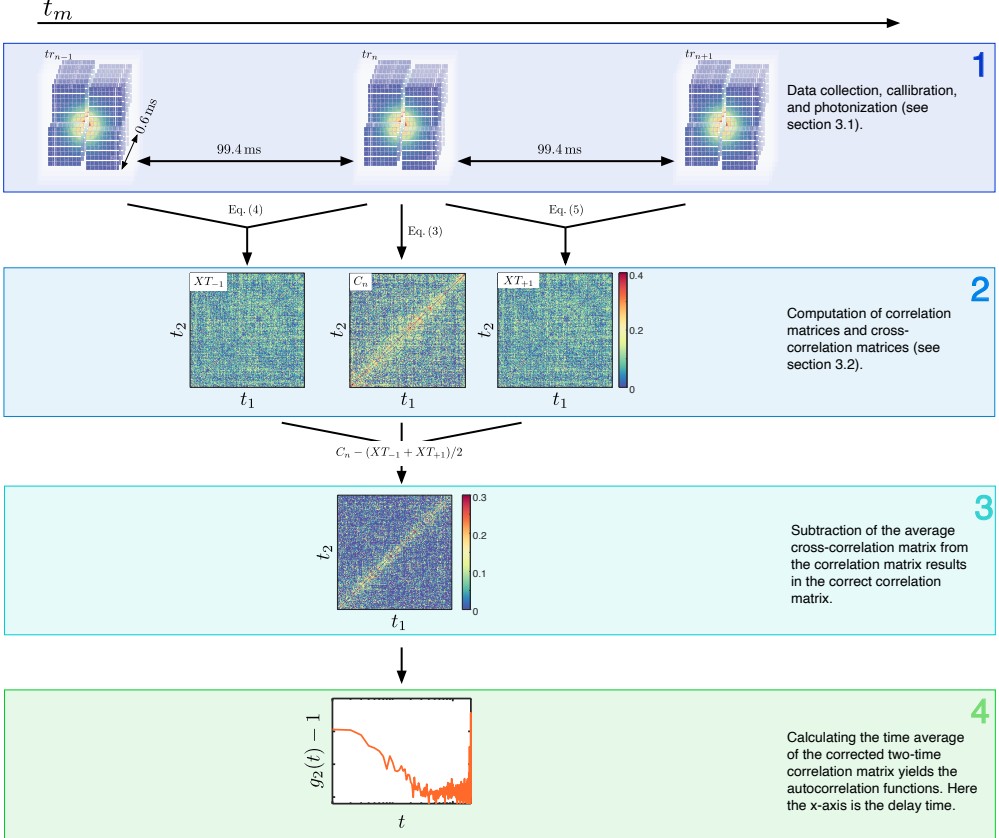

**Figure 1.** Summary of the individual steps of the XPCS analysis pipeline. A short description is given in the figure. More details can be found in the text. We only sketch the analysis of one train for demonstration purposes. Usually, a run contains more than 1000 trains. $t_m$ is the measurement time that can extend over several minutes.

## 2. Materials and Methods

As a model system with well-known dynamic properties, we use silica nanoparticles dispersed in water. The samples were produced such that their intrinsic dynamics match the MHz time scale of the European XFEL. The colloidal silica nanoparticles were synthesized with a modified Stöber method [23]. We used a particle concentration of 2.5 wt% which corresponds to about 1.2 vol%, chosen to ensure a system in which particle–particle interactions are negligible and capable at the same time to provide speckle patterns with sufficient intensity. Monodisperse diluted colloids are extremely useful for calibration purposes because their intermediate scattering function is described by a simple exponential relaxation

$$f(q,t) = \exp(-\Gamma(q)t),\tag{3}$$

where $\Gamma(q) = D_0 q^2$ and $D_0$ is the Stokes–Einstein diffusion constant.

The colloidal dispersions were filled into thin-walled quartz capillaries with an outer diameter of 1.5 mm that were sealed afterwards and placed in a specifically designed sample-holder. The experiment was performed in air at the MID instrument. A detailed description of the instrument can be found in [20].

The intensity of the X-ray pulses was measured on a single-pulse basis with a gas monitor placed upstream the beamline. The beam-size at the sample position was about 10 µm in diameter, obtained by focusing the beam by compound refractive lenses (CRL).

The X-ray flux was controlled with stacks of chemically vapour deposited (CVD) diamond windows. For the data reported here, the total thicknesses ranged from 4.7 mm to 2.7 mm. The X-ray intensity measured by the gas monitor was 1.13 mJ on average. Taking the beamline and air transmission into account, we found that the total attenuation was about $2.44 \times 10^{-4}$ for the data shown here, yielding a fluence on the sample of 3.5 mJ/mm$^2$ per pulse.

The AGIPD installed at MID is composed of four quadrants each consisting of four modules. A module is an array of $2 \times 8$ ASICs (application-specific integrated circuits) where each ASIC is a square of $64 \times 64$ pixels [24] (see Figure 2). The positions of the modules within the same quadrant are fixed, but the relative positions of the four quadrants depend on the configuration. The European XFEL developed a set of software tools that enable an easy geometry definition with the aid of a known reference sample [25]. Once the positions of the direct beam and the modules are defined, it is possible to associate with each pixel the respective magnitude of the exchanged wave vector, $q$. This geometry calibration is fundamental for analyzing the static scattering signal and for defining regions of interest (ROIs) for the XPCS analysis. The intensity is measured by the pixels of AGIPD and is expressed analogous to digital units (ADUs). These are affected by a certain offset, depending on the gain stage in which each individual pixel is operating. Subtracting this offset and calculating the histogram of the data, a red line in Figure 3a, it is possible to identify the ADU value associated with the detection of one or more photons marked as local maxima in the histogram. With this information, it is then possible to perform "photonization" and convert the ADUs to the number of detected photons (see Figure 2).

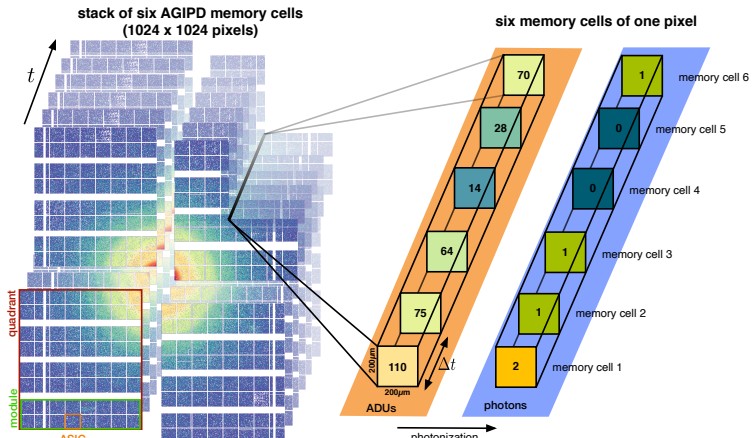

**Figure 2.** Stack of images acquired by the AGIPD. For sake of the visual representation, we limit the figure to six images. In principle, 352 images per train can be acquired with the AGIPD. The data are stored in individual memory cells for each X-ray pulse that arrives at the detector. $t$ is the delay time between successive pulses. In the sketch, the first six memory cells of one pixel are shown. The intensity is measured as analogue to digital units (ADUs) and then converted to the number of photons (photonization, see Figure 3a).

A precise calibration of the AGIPD geometry is achieved with the aid of a previous measurement of the sample at a well-characterized synchrotron beamline. Once the relative positions of the AGIPD modules had been identified, we performed the azimuthal average of the scattered intensity to obtain the $I(q)$ of our sample. In Figure 3b, the comparison between the intensity profiles from this experiment and the one obtained from a measurement performed at beamline P10 of PETRA III is shown, resulting in a sample-detector distance of 7.46 m. The shallow minima in the $I(q)$ from the XFEL data is a consequence of the lower resolution due to the comparably large pixel-size of the AGIPD.

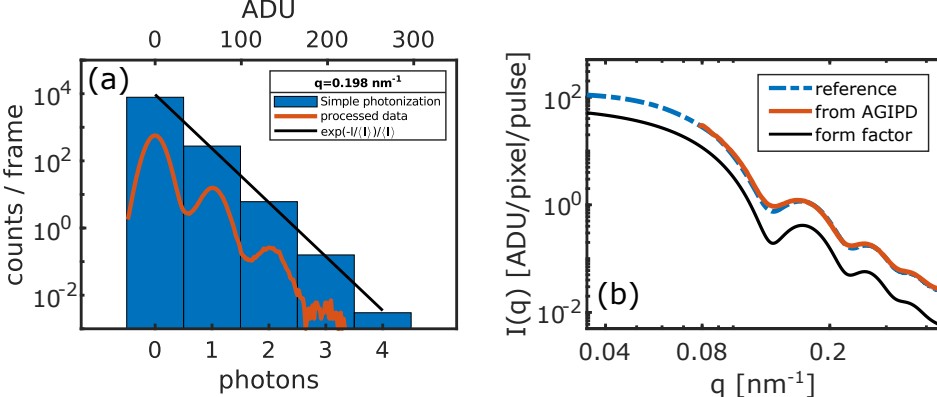

**Figure 3.** (**a**) Histogram of the detected intensity in a $q$-ROI for 1000 frames (red line). By converting the ADUs to number of photons, the blue bar histogram is obtained. The black line indicates the distribution of an ideal speckle pattern. (**b**) $I(q)$ obtained from the AGIPD after the geometry calibration (red line) compared with the $I(q)$ observed from the same sample at beamline P10 (dot-dashed line) measured with an Eiger 4M detector. Taking into account the different resolutions, the average radius is found to be $R_0 = (33.6 \pm 0.1)$ nm and $R_0 = (33.7 \pm 0.1)$ nm for the Eiger 4M and AGIPD, respectively. In black, the form factor of the spherical particles with radius $R_0 = 33.65$ nm and dispersity of 9% is shown (shifted vertically for clarity).

A measurement—at XFEL also denoted a *run*—is gathering data over a certain time $t_{\mathrm{m}}$. During this time, trains of highly intense X-ray pulses arrive at the detector with a separation of 99.4 ms. A train contains currently up to 200 individual X-ray pulses at MID. The delay time between the X-ray pulses within a train can be adjusted in steps of 222 ns which defines the fastest time scale of the correlation function (see Equation (2)).

Each AGIPD module is a separate entity which works independently from the others. To compose an entire scattering pattern, data from the 16 modules need to be grouped. Each module may record a different number of trains which requires a synchronization step to ensure that the final image contains X-rays that arrived at the same time at the detector. This synchronization can be implemented using the train identification number associated with each recorded pattern [25] or using the software *EXtra-data* developed by XFEL [26].

Strictly speaking, Equation (1) only holds for stationary dynamics, i.e., when the dynamics only depend on the delay time but not on the absolute time during the measurement. To measure time-dependent sample dynamics and catch time dependent detector artifacts, we will calculate two-time correlation functions (TTCs) [5,27–30]:

$$C(q, t_1, t_1 + t) = \frac{\langle \delta I(q, t_1) \delta I(q, t_1 + t) \rangle_{sp}}{\langle I(q, t_1) \rangle_{sp} \langle I(q, t_1 + t) \rangle_{sp}}, \tag{4}$$

where $\langle \dots \rangle_{sp}$ denotes the ensemble average over speckles in the same $q$-ROI and $\delta I(q, t_1) = I(q, t_1) - \langle I(q, t_1) \rangle_{sp}$. By the properties of the speckle patterns [31], it is possible to link the two-time correlation matrix to the (square) modulus of the intermediate scattering function, providing information on the density fluctuations and hence internal dynamics of the system that generated the speckle pattern. If the sample is at equilibrium and the dynamics are stationary—within the time-scale of a train—the two-time correlation matrix can be averaged over time $t_1$ yielding $g_2(t) - 1 = \overline{C(t_1, t_1 + t)}$, where $\overline{\phantom{xx}}$ denotes the time average and $g_2(t)$ is the autocorrelation function as introduced in Equation (2). Aiming to investigate sub-microsecond dynamics, we expect that the correlation functions decrease within the time period of one train. Therefore, one TTC per train is calculated and can be seen as an individual measurement. We note that this only holds if the sample is not permanently affected by the X-rays on longer time scales. If beam damage between successive trains is an issue, the sample can be moved continuously so that each train illuminates a new sample volume.

The average over pixels in Equation (4) usually is enough to cancel out any purely random noise sources [28,30], but, in case of the AGIPD (and similar FEL detectors), there are many possible sources of spurious correlations which can generate additional, sometimes time-dependent, terms in the correlation function. The magnitude of these effects can reach comparable or even higher levels with respect to the system's correlation especially at low count rates.

## 3. Results

In this section, we explain the steps of our XPCS data analysis pipeline in detail starting with the data calibration. XPCS experiments typically deal with relatively low intensity signals, since the speckle patters produced by amorphous systems are several orders of magnitude less intense than the Bragg reflections produced by (poly-)crystalline samples. Moreover, to probe intrinsic dynamics, the effect of the X-rays on the sample should be minimized as much as possible within a train, resulting in precise limitations on the incoming X-ray fluence. The practical implications are that all the typical measurements will be realized with the AGIPD in a high gain state, meaning that ideally the measured analogue to digital units (ADUs) linearly increase with the number of photons arriving at the detector.

### 3.1. Corrections to Single Patterns

To clarify the nomenclature: we distinguish between *raw* and *processed* data. Raw data are the ADU values as they are measured by AGIPD and written to disk. The first calibration step is a straightforward pedestal subtraction. This correction step is also called dark subtraction, since the per pixel and per storage cell pedestal is obtained by recording a certain number of trains without X-rays on the detector. This first step is already able to produce reliable results in conditions of fairly good illumination, i.e., close to the centre of the detector, but it quickly fails as soon as the intensity decreases. Therefore, the information obtained from the TTC is tainted by several spurious features as can be seen in Figure 4a. The dynamics in that $q$-region is expected to be described by a fast exponential relaxation with a very mild speeding up along the main diagonal [17], but the measured signal is completely covered by additional contributions and the $g_2(t) - 1$ that can be extracted from such TTC (blue circles in Figure 4d) is completely different from the expected behaviour (black line in Figure 4d).

European XFEL provides a calibration pipeline for calibrating the raw data including many steps in addition to the dark subtraction. A dataset that the XFEL calibration was applied to will be referred to as *processed* data in the following. Processing of AGIPD data has also been tackled by several previous works [10,17,18,32–34]. We will describe here the most important steps included for the data calibration for XPCS data and start with the correction of shifting and fluctuating baseline values. A full detector module's pedestal was found to shift as a function of overall illumination on this module. For that purpose, two Tantalum stripes cover some pixels of each module to measure and subtract this baseline shift. Additionally, the dark pedestal of a single pixel and storage cell may vary over time (see Figure 2), and therefore the recorded signal of a pixel that has not seen any photons can differ from zero even after dark subtraction. It is possible to correct these drifts by subtracting the mean value of empty pixels (identified as having values between −25 ADU and 25 ADU after dark and baseline shift correction). This step is called common mode correction, and it is applied on a per ASIC and per 32 storage cells basis. A last step corrects for different responses of pixels to absorbed photons and multiplies a pre-determined flatfield per pixel and per storage cell. Other corrections can be added, such as removing false intensity gradients across the chips [16,24]. These corrections are implemented in the calibration pipeline provided by European XFEL.

In Figure 4b, the XPCS results obtained with data processed by the XFEL pipeline are shown (dark subtraction, baseline shift correction, common mode, and flatfield are applied). Clearly, the processing of the data solved the issues of the features observed in

the TTC in (a); however, the result is a superposition of a small fast relaxation and a large slow one, also visible in the $g_2(t) - 1$ (red diamonds in Figure 4d). This result is still far from the expected behaviour. To obtain the desired quality of our TTCs, we can implement one drastic but effective measure on top of the data calibration which is "photonization". Analyzing the histograms of the measured ADU values, it is possible to observe that the detection of one 9.3 keV photon produces on average 66 ADUs. Normalizing the data by the single photon value and defining the width of the bins, we can identify how many photons were recorded in every pixel and memory cell. The size of the bin will determine how selective our photonization procedure will be. Depending on the photon energy, which defines the separation of the photon events in the histogram, and required precision, one can adjust the bin size. In the present case, since the distance between the photon peaks is on the order of $\sim 2\sigma_{ph}$, where $\sigma_{ph}$ is the width of one peak, a simple rounding operation is sufficient. Most of the positive effects of the photonization originate from the fact that all pixels without photons are set to 0. With XPCS, typical count rates are of $1 \times 10^{-2}$ to $1 \times 10^{-1}$ photons per pixel, thus the number of pixels detecting 0 photons are several orders of magnitude larger than the number of pixels that counted one or more photons, as can also be seen from the histograms in Figure 3a. Performing the correlation of the photonized dataset, the TTC in Figure 4c is obtained, where it is possible to finally observe the expected qualitative behavior. The necessity of the photonization step is evident in particular in Figure 4d where we compare the resulting $g_2$-functions after time averaging the TTCs obtained from different algorithms. The results after photonization are the closest to the theoretical predication. This benchmark clearly shows that, without the correct analysis scheme, the correlation functions do not reflect the sample dynamics but are dominated by detector artifacts and possibly other sources of fluctuations.

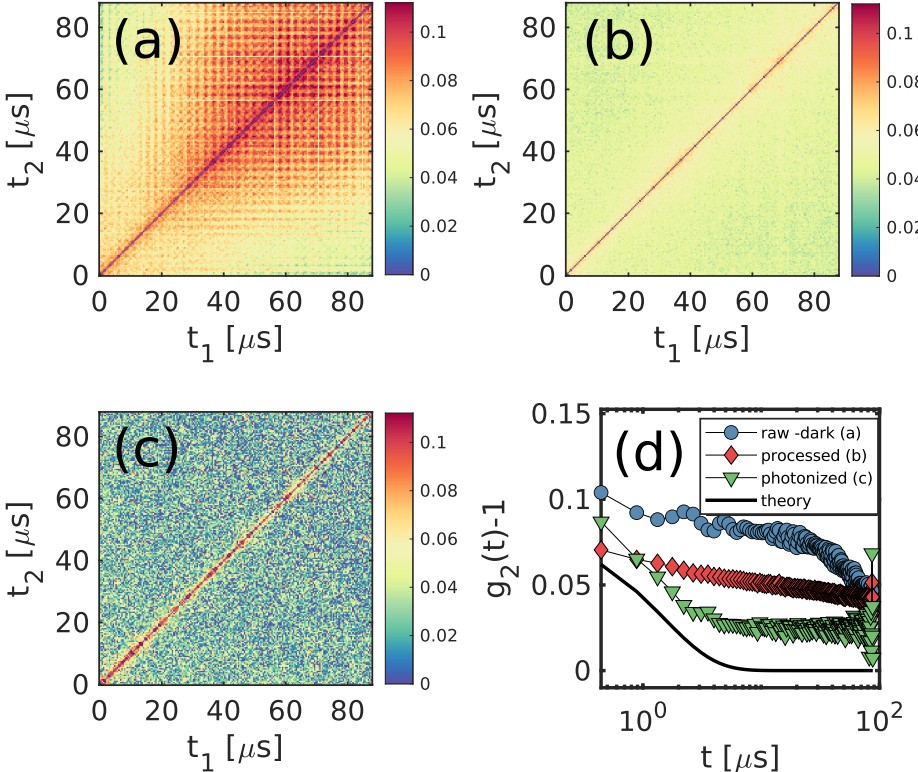

**Figure 4.** Averaged TTC obtained from different analysis pipelines applied to the same dataset at $q = 0.203$ nm$^{-1}$: (**a**) basic background subtraction; (**b**) XFEL correction pipeline; and (**c**) photonized data. The $g_2(t) - 1$ functions from the TTC (**a–c**) are reported in (**d**) as blue circles, red diamonds, and green triangles, respectively. A black continuous line indicates the expected behaviour, for a $g_2$ function with a contrast of $\beta_0 = 0.08$.

### 3.2. Corrections on the Dynamical Quantities

Despite of the correction of single patterns, we have not yet reached an optimal result, as the value of the correlation at large lag times, visible also from the $g_2(t) - 1$ in Figure 4d (green triangles), is still well above 0. This additional baseline originates from the static variance of the recorded intensity in the investigated $q$-ROI. It can have several origins, such as the natural change of the intensity as a function of $q$, small misalignments of the modules or other artifacts which have yet to be clearly identified. However, this non-zero baseline is not the only concern; in low-intensity regions, the data are much more exposed to the appearance of "hot pixels" or other similar defects, which usually affect the TTC with strongly correlated streaks or blocks. Figure 5a reports an example of such artefacts, with a typical block of 32 memory cells that shows a very high, and unphysical, correlation. For clarification, this does not mean that all those memory cells are affected, but only the memory cell of a sub-group of the pixels (in most cases, only a single pixel) in the investigated $q$-ROI. On conventional detectors, it would be possible to identify and mask such pixels, but, in the present case, such effects can also be observed on different memory cells on different runs, suggesting a variability which is not yet completely under control. Luckily, there are methods to tackle this problem thanks to the cross-correlation matrices introduced in the previous section. In fact, correlating neighbouring trains, a precise estimation of such a baseline can be obtained because illumination and detector conditions are very similar, but the speckle patterns are completely uncorrelated. For a given train $tr_n$ then, two cross-correlation matrices are calculated as

$$X_{-1} = \langle \delta I(tr_{n-1}, t_1) \delta I(tr_n, t_2) \rangle / (I(tr_{n-1}, t_1) I(tr_n, t_2)) \tag{5}$$

between $tr_{n-1}$ and $tr_n$, and

$$X_{+1} = \langle \delta I(tr_n, t_1) \delta I(tr_{n+1}, t_2) \rangle / (I(tr_n, t_1) I(tr_{n+1}, t_2)) \tag{6}$$

between $tr_{n+1}$ and $tr_n$. The cross-correlations are then made symmetric with $XT_{-1} = (X_{-1} + X_{-1}^T)/2$ and finally the correction matrix is defined by

$$XC = (XT_{-1} + XT_{+1})/2. \tag{7}$$

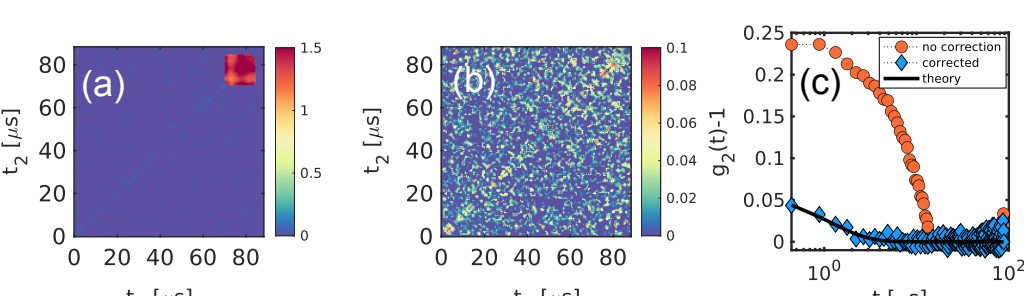

**Figure 5.** Effects of the correction scheme on low-intensity $q$-regions, $q = 0.233$ nm$^{-1}$ in this example. (**a**) TTC from a photonized dataset without applying further corrections. A typical artefact of the AGIPD represented by a group of 32 neighbouring memory cells that display a non-physical high correlation is highlighted (red square in the upper right corner); (**b**) the same TTC after the cross-correlation correction; (**c**) autocorrelation functions obtained from the TTC (**a**,**b**) compared with the expected behaviour (black line).

Finally, simply subtracting $XC$ from the TTC corrects the baseline and some of the more common artefacts [27], as can be seen in the corrected TTC shown in Figure 5b. To obtain a more quantitative insight on the effects of this correction in Figure 5c, the $g_2(t)$ functions calculated from uncorrected and corrected TTC are displayed. There it is possible to notice how strongly the presence of the "wrong" memory cells influence the signal showing a large relaxation where the expected dynamics would be already almost

completely decorrelated. It must be pointed out that this correction scheme is based on two conditions: the speckle patterns in two different trains are uncorrelated and the azimuthal average of $I(q, \tau)$ is the same in all trains. In other words, each train probes a reproducible condition and is a statistically independent measurement of the dynamics.

### 3.3. Benchmark on Diffusing Samples

The correction schemes presented so far can provide reasonable results, but, to certify that such results are correct, we must benchmark them with theoretical predictions. The $g_2$ functions obtained performing only the dark subtraction, photonization, and the cross correlation are reported in Figure 6a. For comparison, the $g_2$ functions obtained from the same data but with a pipeline that performs all the calibration steps except the "photonization" are shown in Figure 6b. The correlation functions at the lower $q$-values are almost identical, and they become more different for larger $q$. Not only does the contrast in Figure 6b drop more abruptly, but the time-scale of the relaxation is also different. A more quantitative comparison can be done by performing an exponential fit to the data (black lines) and plotting the respective relaxation rates. In Figure 6c, $\Gamma(q)$ obtained from these two different sets of functions are shown. The value of $D_0 = (8.1 \pm 0.6)$ nm$^2/\mu$s is obtained from a fit of $\Gamma(q)$ restricted to the values at q lower than 0.113 nm$^{-1}$, where both pipelines provide the same results, highlighted by filled symbols in the figure. $D_0$ can be also obtained from the the Stokes–Einstein relation for spherical particles such as $D_0 = k_B T / (6\pi \eta(T) R_0)$, with $k_B$ Boltzman's constant, $T$ the temperature, and $\eta(T)$ the solvent's viscosity. Considering the average radius $R_0$ obtained from the fits of the $I(q)$ (Figure 3), we find the same diffusion constant for temperatures $T \sim 302$ k, as expected from the fluence and repetition rate used for the measurement [17,18]. In our case, the predictable behaviour of the simple diffusion is useful to have a clear view of the differences between the results of the two pipelines. Extrapolating the $D_0 q^2$ law to the whole probed $q$-range (black dashed line in Figure 6c), we can clearly see that the results from the "photonized" (blue data) patterns remain close to the expected result, while the other (red data) strongly deviate in a non-monotonic manner, indicating that the dynamics results are dominated by the artifacts that have been reported e.g., in Figure 4b.

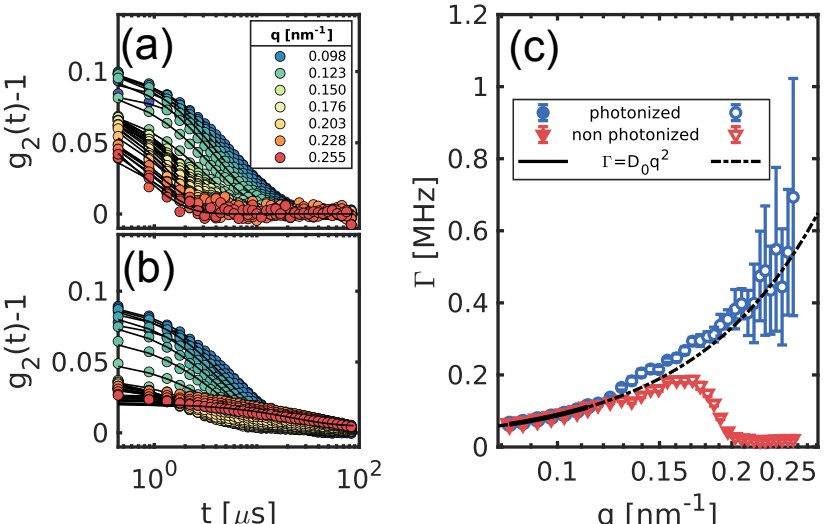

**Figure 6.** Practical example of the advantages offered by the "photonization" of the speckle patterns. In (**a**), the $g_2(t) - 1$ obtained from the photonized patterns are reported along with their respective exponential fits (black lines). For comparison, in (**b**), the $g_2(t) - 1$ from the same run, but without the photonization step are reported. In (**c**), the relaxation rates obtained with simple exponential fits to the data are shown. The data points used for the fit of $D_0$ are marked as filled symbols, the black continuous line is the fitted function, while the black dash dotted line is the extrapolation to the whole q-range.

## 4. Discussion

The situation presented in the previous paragraph is a straightforward example of the advantages that come with the "photonization" step that eliminates correlated noisy pixels. In sequential XPCS, the choice of the criteria that determine the identification of a photon is not very critical since, as long as the mistaken photons are purely random, they will contribute to the correlation functions only for $t = 0$. Conversely, the choice of the parameters for the photonization is of primary importance for other classes of correlation experiments such as XSVS [35], where the contrast from single frames is used to evaluate the dynamics at very short timescales and the average number of photons per pixels is typically very low ($<10^{-2}$). In this case, more sophisticated methods of converting the measured ADU values to photons might be necessary. That simple thresholding—the method we use here—might fail to unambiguously identify a photon. This can be seen from the histogram of Figure 3a, where the noise and the photon peaks partially overlap. Thus, it is not possible to univocally attribute the number of photons for a certain fraction of events. For detectors with smaller pixels than the AGIPD, one photon might create charges in several adjacent pixels. Then, dropletization algorithms [36] can be used to recombine the ADU values to photons and assign the correct number of photons to each pixel. In case of AGIPD, the large pixel size limits the spatial cross-talking between pixels and dropletization becomes equivalent to photonization. In the future, a special measuring mode (high-CDS) will be available for MID users, which is supposed to increase the separation of the photon peaks and the accuracy of the photonization. Another solution would be to increase the energy of the photons which would also increase the distance between the photon peaks. However, this approach comes with a drawback: a higher energy implies more possibilities to excite fluorescence in some elements of the beamline or the sample and the intensity of the speckle patterns will be weaker for the same fluxes. Furthermore, the speckle size will be smaller, which will reduce the contrast and the signal-to-noise ratio. In general, for the outcome of a very low intensity experiment, any method that can make the AGIPD work with a precise single photon resolution, either an even more refined calibration pipeline or different experimental conditions, should be pursued.

In this paper, we showed the key aspects of sequential XPCS experiments that make use of the unique pulse structure of the European XFEL. We outlined the unique technical details of this class of experiments, and the general steps that should be considered for a successful data analysis. We tested different calibration pipelines on a prototypical sample composed of diluted nanoparticles dispersed in water. Among the various steps that can be taken to process the data, we identified two steps that provide the largest effect for the determination of dynamical quantities: (i) photonization of the speckle pattern that expresses the values in the recorded frames as integer numbers and (ii) correction of the TTC by the cross-correlation matrices obtained from neighbouring trains. Lastly, we outlined the limitations of the current pipelines which might have severe consequences on other photon correlation techniques and we proposed some possible solutions.

**Author Contributions:** Conceptualization, F.D., M.R. and F.L.; Software, F.D., M.R. and A.J.; Formal Analysis, F.D. and M.R.; Investigation, F.D., M.R., I.L., J.M., M.S., A.M., F.P. and F.L.; Resources, I.L.; Supervision, G.G., F.P. and F.L.; Writing—Original Draft Preparation, F.D., M.R. and F.L.; Writing—Review and Editing, all authors. All authors have read and agreed to the published version of the manuscript.

**Funding:** This research is supported by the Cluster of Excellence "Advanced Imaging of Matter" of the Deutsche Forschungsgemeinschaft (DFG)—EXC 2056—project ID 39071599—and by the Centre for Molecular Water Science (CMWS) in an Early Science Project. We acknowledge financial support by the Swedish National Research Council (Vetenskapsrådet) within the Röntgen-Ångström Cluster Grant No. 2019-06075.

**Institutional Review Board Statement:** Not applicable.

**Informed Consent Statement:** Not applicable.

**Data Availability Statement:** The data presented in this study are available on request from the corresponding authors. Raw data will become available 2023 via doi:10.22003/XFEL.EU-DATA-002616-00.

**Acknowledgments:** We acknowledge European XFEL in Schenefeld, Germany, for provision of X-ray free electron laser beamtime at Scientific Instrument MID (Materials Imaging and Dynamics) and would like to thank the staff for their assistance. We acknowledge DESY (Hamburg, Germany), a member of the Helmholtz Association HGF, for the provision of experimental facilities. Parts of this research was carried out at PETRA III, and we would like to thank Michael Sprung and Fabian Westermeier for assistance in using beamline P10. We thank Kartik Ayyer, Ulrike Boesenberg, Steffen Hauf, and Jolanta Sztuk–Dambietz for discussion on AGIPD data corrections and Wonhyuk Jo for comments on the manuscript.

**Conflicts of Interest:** The authors declare no conflict of interest.

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
