# Peer review of "Analysis Strategies for MHz XPCS at the European XFEL"

_applsci, doi:10.3390/app11178037_

Round 1

Reviewer 1 Report

X-ray Photon Correlation Spectroscopy has been used at synchrotron radiation facilities for studying dynamics in the time range between hours and milliseconds. Recent advance in X-ray Free Electron Laser makes it possible to study dynamics in femtoseconds to sub-nanosecond time-scale. The authors have developed a methodology to fill the gap of time scales between nanoseconds to milliseconds by utilizing a unique pulse structure available at the European XFEL. Although useful to fill the gap, this unique pulse structure poses significant challenges regarding detectors, data handling, and analyses. The current manuscript describes these challenges and solutions to overcome them in detail. I found some typos and grammatical errors, but overall the manuscript is well written. The reviewer recommends the publication of this manuscript. 

Author Response

We thank the reviewer for the positive feedback. We carefully proofread the manuscript and corrected typos and grammar.

Reviewer 2 Report

The manuscript presents the analysis strategy for sequential mode XPCS measurements that utilize the pulse train structure of the European XFEL. The pipeline for extracting the time correlation functions in the speckle patterns recorded using the megapixel AGIPD detector was demonstrated with a well-known model system – dilute silica nanoparticles in water.

The paper provides a detailed description of systematic errors in correlation function extraction due to the detector artifacts and highlights the methods to correct for them, which will be very beneficial for user experiments not only in the MID instrument, but across all x-ray FEL facilities for sequential mode XPCS measurements. I therefore support the publication of this article in Applied Sciences. To increase the readability of the manuscript, below are a few questions that need to be clarified.

  1. Line 100-108 describes the pulse energy measurement; the paper should then provide the averaged pulse energy at the sample.
  2. The setup geometry: 9.3keV, 10-micron beam, detector sample distance ~7.46 meter, 200 micron detector pixel size, this gives a speckle size below half a pixel size. Is there a specific reason to measure at the under-sampled condition?
  3. Following the previous question, the contrast seems quite low to start with, this can be partly attributed to detector under sampling. On the other hand, is there a monochromator used for the measurement and what is the sample thickness? What will the general strategy of measuring in the wide angle where the count rate and contrast will be even lower?
  4. Figure 3 (b) gives the measured I(q).
    • It would be informative to show that in photons/pixel/pulse rather than arbitrary unit as the low count rate in the higher q regions might account for the biased contrast extraction in section 3.3.
    • The deviation from the ideal form factor is highly likely due to a size distribution in the sample size. I suggest fitting to the experimental data considering the distribution as it will give us the error bar estimation for the theoretical diffusion coefficient (D0).
  5. The histogram (Fig.3 (a)) should give a good estimation of the charge cloud size, how big is it in the unit of detector pixel? The photonization process is simply performed by rounding photons in the manuscript, what is the estimated error in correlation?
  6. Regarding section 3.2, for correcting correlation matrices. Mathematically speaking, should it be performed by subtraction or division((Cn+1)/(XC+1))? The detector artifact contributes to the correlation (g2a) in addition to that from the sample dynamics (g2s). Then the overall g2 = g2a* g2s since the two are independent.
